# The Effects of the Food Additive Titanium Dioxide (E171) on Tumor Formation and Gene Expression in the Colon of a Transgenic Mouse Model for Colorectal Cancer

**DOI:** 10.3390/nano12081256

**Published:** 2022-04-07

**Authors:** Nicolaj S. Bischoff, Héloïse Proquin, Marlon J. Jetten, Yannick Schrooders, Marloes C. M. Jonkhout, Jacco J. Briedé, Simone G. van Breda, Danyel G. J. Jennen, Estefany I. Medina-Reyes, Norma L. Delgado-Buenrostro, Yolanda I. Chirino, Henk van Loveren, Theo M. de Kok

**Affiliations:** 1Department of Toxicogenomics, GROW School for Oncology and Reproduction, Maastricht University Medical Center, 6229 ER Maastricht, The Netherlands; heloise.proquin@rivm.nl (H.P.); marlon.jetten@maastrichtuniversity.nl (M.J.J.); yannickschrooders@xs4all.nl (Y.S.); marloes.jonkhout@kuleuven.be (M.C.M.J.); j.briede@maastrichtuniversity.nl (J.J.B.); s.vanbreda@maastrichtuniversity.nl (S.G.v.B.); danyel.jennen@maastrichtuniversity.nl (D.G.J.J.); henk.van.loveren@gmail.com (H.v.L.); t.dekok@maastrichtuniversity.nl (T.M.d.K.); 2National Institute for Public Health and Environment (RIVM), Bilthoven, 3721 MA De Bilt, The Netherlands; 3Faculty of Health, Medicine and Life Science, Maastricht University Medical Center, 6229 ES Maastricht, The Netherlands; 4Laboratory of Biosignaling & Therapeutics, Department of Cellular and Molecular Medicine, KU Leuven, 3000 Leuven, Belgium; 5Laboratorio de Carcinogénesis y Toxicología, Unidad de Biomedicina, Facultad de Estudios Superiores Iztacala, Universidad Nacional Autónoma de México, Mexico City 54090, Mexico; medinaingrid0@gmail.com (E.I.M.-R.); nlbuenrostro@gmail.com (N.L.D.-B.); irasemachirino@gmail.com (Y.I.C.)

**Keywords:** titanium dioxide, E171, mice, transgenic, tumor formation, gene expression, toxicology, in vivo

## Abstract

Titanium dioxide (TiO_2_) is present in many different food products as the food additive E171, which is currently scrutinized due to its potential adverse effects, including the stimulation of tumor formation in the gastrointestinal tract. We developed a transgenic mouse model to examine the effects of E171 on colorectal cancer (CRC), using the Cre-LoxP system to create an *Apc*-gene-knockout model which spontaneously develops colorectal tumors. A pilot study showed that E171 exposed mice developed colorectal adenocarcinomas, which were accompanied by enhanced hyperplasia in epithelial cells, and increased tumor size. In the main study, tumor formation was studied following the exposure to 5 mg/kg_bw_/day of E171 for 9 weeks (Phase I). E171 exposure showed a statistically nonsignificant increase in the number of colorectal tumors in these transgenic mice, as well as a statistically nonsignificant increase in the average number of mice with tumors. Gene expression changes in the colon were analyzed after exposure to 1, 2, and 5 mg/kg_bw_/day of E171 for 2, 7, 14, and 21 days (Phase II). Whole-genome mRNA analysis revealed the modulation of genes in pathways involved in the regulation of gene expression, cell cycle, post-translational modification, nuclear receptor signaling, and circadian rhythm. The processes associated with these genes might be involved in the enhanced tumor formation and suggest that E171 may contribute to tumor formation and progression by modulation of events related to inflammation, activation of immune responses, cell cycle, and cancer signaling.

## 1. Introduction

Colorectal cancer (CRC) is a complex disease with high morbidity and mortality that is persistent in Western countries, which displays an increasing risk to the younger population [1,2]. Dietary factors are known to be involved in the development of CRC [3] and small particles, such as nanoparticles (<100 nm), in food are suspected to induce adverse effects due to their size-dependent reactivity [4,5].

Titanium dioxide (TiO_2_) is a food additive that has been approved in food products by the European Union (EU), coded E171. It consists of at least 10–40% nanoparticles in number–size distribution, with one or more external dimensions in the size range of 1–100 nm [5,6,7]. E171 is used for its bright and white color, high refraction index, and resistance to UV light, which makes it a very stable pigment over time. Consequently, E171 is used in a wide variety of foods, such as dairy products, sweets, cookies, and sauces [5,8,9]. The adverse effects of TiO_2_ were first observed after inhalation. As a consequence of these findings, the International Agency for Research and Cancer (IARC) has classified TiO_2_ as “possibly carcinogenic to humans after inhalation” [10]. In 2020, the EU also classified TiO_2_ as a suspected carcinogen (category 2) by inhalation in powder form with at least one particle dimension with an aerodynamic diameter ≤10 µm under the Classification, Labelling, and Packaging (CLP) Regulation (EC No 1272/2008) [11]. This classification was mandatory from 1 October 2021 [11]. Findings in inhalation studies as well as the change of classification to group 2B raised concern about other routes of exposure, such as ingestion [10,11].

Therefore, many studies have been performed to identify the potential carcinogenicity of E171 after ingestion, as well as other related mechanisms. Based on the outcome of these studies, as of May 2021, E171 is “no longer considered to be safe when used as a food additive” by the European Food Safety Authority (EFSA), particularly due to concerns about its genotoxic potential [5].

The potential of TiO_2_ to enhance CRC has been shown in different rodent models. In a chemically induced CRC model, based on the combination of azoxymethane (AOM) and dextran sodium sulfate (DSS), BALB/c mice were orally exposed to 5 mg/kg_bw_/day of E171 for 10 weeks [12]. The expression of tumor progression markers, such as COX2, Ki67, and ß-catenin, was increased [12]. In the same study, no development of tumors was observed after the exposure to E171 alone; although, a decrease in goblet cell numbers and induction of dysplastic changes in the colonic epithelium were detected [12]. Another study in a chemically induced CRC model showed the growth of aberrant crypt foci in the colon after oral exposure to 10 mg/kg_bw_/day of E171 for 100 days [13]. Bettini et al. (2017) also showed the development of preneoplastic lesions in normal rats after oral exposure to 10 mg/kg_bw_/day of E171 for 100 days [13]. They found a shift in T-helper cell 1/T-helper cell 17 (Th1/Th17) balance in the immune system and observed impaired intestinal homeostasis in rats exposed to E171 for 7 days [13]. Gene expression analysis after the oral exposure to 5 mg/kg_bw_/day of E171 for 2, 7, 14, and 21 days in the colon of normal BALB/c mice showed changes in olfactory/G-protein-coupled receptor (GPCR) signaling genes, immune system, oxidative stress responses, and cancer-related genes [14]. Similar results were observed in a colitis-induced mouse model (AOM/DSS) with the same experimental design. Gene expression changes in the colon of these mice indicated modulations of immune-related genes, olfactory/GPCR signaling genes, oxidative stress, extracellular matrix modulation, and biotransformation of xenobiotics [15].

In the current study, we used a transgenic (Tg) mouse model based on the Cre-LoxP system to produce a tissue-specific knockout model for CRC, which does not require chemical induction of inflammatory colorectal cancer [16]. The Cre-LoxP system is a tissue-specific gene-editing technology, which allows researchers to carry out deletions, insertions, and translocations in a site-specific manner [17]. A knockout of the floxed *Apc* gene in the colon by Cre-recombinase with the *Car1* promoter induced 26% of the mice to spontaneously develop colorectal tumors, which were limited to the epithelial of the large bowel [16]. The model is representative of the human situation, since the loss of function of the *Apc* gene after mutation, and the resulting transformation of the normal epithelium to early adenoma/dysplastic crypts, is suspected to be the primary cause of sporadic and hereditary CRC and is found in around 80% of human colorectal tumors [18]. To the best of our knowledge, this study is the first to utilize this transgenic mouse model for CRC to screen for effects on tumorigenesis and transcriptome changes following exposure to the food additive E171.

We investigated the effects of intragastric exposure to E171 in the distal colon of this Tg mouse model by monitoring tumor formation and gene expression profiling following a similar exposure schedule to that previously reported [14,15]. We hypothesize that E171 induces gene expression changes that may lead to altered signal transduction, oxidative stress response, inflammation, impairment of the immune system, and the activation of cancer-related genes, which may stimulate colorectal tumor formation.

## 2. Materials and Methods

### 2.1. E171 Characterization

E171 was kindly donated by the Sensient Technologies Company in Mexico. The characterization of E171 by electron microscopy was used to evaluate the size morphology. In short, E171 was dispersed in sterile water with a bath sonicator (Branson 2200, Branson Ultrasonic SA, Danbury, CT, USA) at 40 kHz for 30 min (unless stated otherwise). The vehicle control, consisting of sterile water, was also bath-sonicated for 30 min at 40 kHz. E171 primary particle size was analyzed by transmission electron microscopy (TEM), with a FEI Tecnai G2 Spirit BioTWIN electron microscope at 60,000× magnification (Thermo Fisher Scientific, Hillsboro, OR, USA). A measure of 20 µL of E171 dispersion in sterile water was added to an EMS CF200-Cu-50 support film for EM (Electron Microscopy Science, Hatfield, PA, USA). For each E171 concentration, 50 pictures were taken and analyzed using the ellipse fitting mode of the ParticleSizer plug-in for ImageJ, which was developed by the NanoDefine project [19,20]. At least 1500 particles were analyzed for each sample. A Malvern NanoZS (Malvern Instruments, Malvern, UK) dynamic light scattering instrument was used to determine hydrodynamic diameter size and zeta-potential The stock dispersions of 1, 2, and 5 mg/mL were diluted 1:100 (*v/v*). Measurements were performed as biological duplicates at 25 °C, with equilibration time of 0 s, viscosity of 0.8872 cP, and the refraction index was set at 1.330.

### 2.2. Animals

All animal experiments were reviewed and approved by the Animal Experimental Committee, which included an ethical testing framework from Maastricht University, Maastricht, The Netherlands, or the ethics committee of Facultad de Estudios Superiores Iztacala from Universidad Nacional Autónoma de México, Mexico City, Mexico. The pilot study was approved under the number CE/FESI/102021/1430. The phase I tumor formation study was approved under the number 2014-080 and the phase II gene expression study under the number 2014-079. All mice were housed in polycarbonate cages and kept in a housing room (21 °C, 50–60% relative humidity, 12 h light/dark cycles, air filtered until 5 µm particles and exchanged 18 times/h). Mice received a standard chow diet and water ad libitum. Cardboard was removed from cages to avoid potential contamination with TiO_2_ [21]. The doses of 1, 2, and 5 mg/kg_bw_/day of E171 were based on the EFSA exposure assessment estimation of the mean dietary intake of adults (18–64 years) [5]. The CAC5^Tg/Tg^ expressing the Cre recombinase under the control of the *Car1* promoter and the APC^580S/+^ mice carrying the floxed *Apc* gene were crossed to obtain CAC^Tg/Tg^;APC^580S/+^ mice based on the model created by Xue et al. [16]. The combination of LoxP and Cre-recombinase, expressed in the colon, induced a spontaneous development of colon carcinomas in 26% of the mice within the first 10 weeks of age [16]. Heterozygosis of the *Apc* gene and the homozygosis of Cre-recombinase with the Car-1 promoter was confirmed by Charles River Biopharmaceutical Service GmbH, Germany, by extracting DNA from the tip of the tails of the mice, to perform a polymerase chain reaction (PCR) before the start of the phase I and phase II experiments.

### 2.3. Pilot Study: Histopathology

A pilot study was performed to investigate whether the Tg mice developed tumors and whether E171 would stimulate the development of colorectal tumors. In this pilot experiment, 3 heterozygous CAC^Tg/Tg^; APC^580S/+^ mice were treated with ~5 mg/kg_bw_/day of E171 via drinking water for 4 weeks, 7 days a week. Another 2 mice served as control receiving only sterile water. The mice were exposed from 35 weeks of age. The water container was filled with 200 mL of water or 200 mL of 22 µg/mL E171 dispersion (bath sonicated at 60 kHz for 30 min at room temperature) every 2 days and was shaken at least 2 times per day. Water consumption was monitored. Euthanasia was performed under light anesthesia (xylazine/ketamine 10/80 mg/kg_bw_) followed by cervical dislocation. Colorectal tumors were counted, measured, and histological samples were taken. The whole colon and the cecum were isolated and photographed (data not shown) before the cecum was removed and the colon was washed with a cold saline solution. Then, the colon was dissected lengthwise and photographed (data not shown). The length and width of the adenomas were measured using digital calipers and the volume (mm^3^) was calculated considering ((LxW^2^)/2), where L indicates length and W indicates width. Then, the colon was fixed in 100% alcohol for 24 h, dehydrated, and embedded in paraffin. Tissue sections of 4 µm were obtained from the paraffin-embedded samples and stained. Briefly, slides were hydrated by washes with xylene, xylene/ethanol, 100% ethanol, 96% ethanol, and water. The slides were routinely stained with hematoxylin and eosin (HE). The samples were dehydrated and preserved with Entellan^®^. The histopathology was determined by a certified histopathologist.

### 2.4. Phase I: Tumor Formation Study

The 80 CAC^Tg/Tg^;APC^580S/+^ pups were weaned at 3 weeks of age. Mice were randomly divided into two groups: the E171 exposure group and the control group. From 5 weeks of age, the CAC^Tg/Tg^;APC^580S/+^ mice were treated with 5 mg/kg_bw_/day of E171 (n = 40) or sterile water (n = 40) for a maximum of 9 weeks. Mice were given 250 µL of E171 dispersion or sterile water by intragastric administration. The number of particles in 250 µL of a 1 mg/mL stock dispersion was ~2.3 × 10^11^ (based on a median of ~79 nm). Exposure was performed 5 days a week. Weight gain or loss was monitored weekly. A total of 16 mice, with 8 in the E171 group and 8 in the control group—4 males and 4 females in each group—were euthanized bi-weekly. Euthanasia was performed under light anesthesia (4% isoflurane) followed by cervical dislocation. At week 4 of exposure, one mouse in the E171 group (group of 9 weeks of exposure) was euthanized ahead of schedule due to severe rectal prolapse. At week 7 of exposure, one more mouse in the control group had tumors all over the colon from the caecum to the distal colon, which is unusual in this model [16]. Therefore, this mouse was considered an outlier and was not included in the assessment of the tumor formation. Colon, liver, and spleen were removed and weighed. Colons were cleaned with a sterile swab before they were weighed and checked for the presence of tumors; the number of visible tumors was registered.

### 2.5. Phase II: Gene Expression Study

For the gene expression study, 112 CAC^Tg/Tg^;APC^580S/+^ mice were randomly divided into 4 groups: 3 exposure groups with different concentrations of E171 (1, 2, and 5 mg/kg_bw_/day) and 1 control group exposed to sterile water. The mice, at 5 weeks of age, were exposed 5 days a week, as indicated in the tumor formation study. After 2, 7, 14, and 21 days of exposure, 28 mice were euthanized: 7 mice per group, with 3 females and 4 males in each group. Three mice were euthanized ahead of schedule due to severe rectal prolapse and were not included in further analysis. There was 1 mouse in the 1 mg/kg_bw_/day of E171 group of 21 days of exposure, and 2 mice in the control group, with 1 of the 14 days group and 1 of the 21 days group. Colon, liver, and spleen were sampled from every mouse. The presence of tumors in the colon was registered. Colons were cleaned with a sterile swab before they were weighed and dissected into the distal, mid, and proximal colon. One segment of each dissection was stored at −80 °C in RNAlater (Thermo Fisher, Breda, The Netherlands).

### 2.6. mRNA Extraction from Colonic Tissues

As tumor formation in this mouse model was mainly found in the distal colon [16], mRNA was extracted from this part of the colon as previously reported [14,15]. Briefly, before RNA isolation, the distal colon was disrupted and homogenized by submerging it in Qiazol (Qiagen, Venlo, The Netherland) and by subsequently using a Mini Bead Beater (BioSpec Products, Bartlesville, OK, USA) at a speed of 48 beats per second for 30 s. mRNA isolation followed the manufacturer’s protocol for “Animal Cells and Animal Tissues” in the mRNAeasy Mini Kit (Qiagen, Venlo, The Netherlands), with DNase treatment (Qiagen, Venlo, The Netherlands) [22]. The quality and yield of the mRNA were verified on a Nanodrop^®^ ND-1000 spectrophotometer (Thermo Fisher, Breda, The Netherlands). Samples passing the quality check with a 260/230 ratio between 1.8–2.0 and a 360/280 ratio between 1.9–2.1 were checked for RNA integrity on a 2100 Bioanalyzer using the manufacturer’s protocol (Agilent Technologies, Amstelveen, The Netherlands). All samples with an RNA integrity number (RIN) above 6 were used for the microarray analysis. The average RIN value of the 99 samples that were used was 7.8 ± 0.8.

### 2.7. cRNA Synthesis, Labeling, and Hybridization

Samples were prepared for microarray analysis by synthesizing the mRNA into cRNA, labeling with Cy3, amplifying, and purifying it using the RNeasy Mini Kit (Qiagen, Venlo, The Netherlands) according to the One-Color Microarray-Based Gene Expression Analysis Protocol Version 6.6 [23]. Furthermore, quantification of Cy3 labeled to the cRNA was performed by using a Nanodrop^®^ ND-1000 spectrophotometer with microarray measurement. For hybridization, 600 ng of labeled cRNA was used on SurePrint G3 Mouse Gene Expression v2 8 × 60K Microarray slides (Agilent Technologies, Amstelveen, The Netherlands). Moreover, microarray slides were scanned using an Agilent DNA Microarray Scanner with Surescan High-resolution Technology (Agilent Technologies, Amstelveen, The Netherlands). The scanner was set to Dye Channel: G, Profil: AgilentG3_GX_1Color, Scan region: Agilent HD (61 × 21.3 mm), scan resolution of 3 µm, Tiff file dynamic range of 20 bit, red PMT gain of 100%, and green PMT gain of 100%.

### 2.8. Preprocessing and Data Analysis of Microarrays

Preprocessing of microarray raw data was performed as previously described [14]. Briefly, gene expression values were obtained from the microarray scans using Agilent’s Feature Extraction software (FES) v10.7.3.1). Next, the samples were checked for quality and normalized with ArrayQC, an in-house developed pipeline (https://github.com/arrayanalysis/arrayQC_Module, accessed on 1 February 2021) using the following steps: local background correction, flagging of bad spots, controls/spots point with too low intensity, log2 transformation, and quantile normalization. Bad spots were removed from the normalized data (normalized data can be accessed on GEO under the number GSE186727). A total of 16 groups were defined, based on exposure and timepoints: 4 different exposure groups for 4 different timepoints. Spot identifiers were deleted when more than 40% of samples in each group had a missing value and when the average expression in each group was <4. Missing values were imputed by the k-nearest neighbors using GenePattern ImputeMissingValues. KNN module v13 (k-value 15) [24]. Repeated Agilent probe identifiers were merged (setting: median) with Babelomics 5. Next, Agilent probe identifiers were reannotated to EntrezGeneIDs and again merged (setting: median) with Babelomics 5 [25]. Differentially expressed genes (DEGs) were identified by performing a moderate *t*-test using LIMMA (v1.0), which corrected the exposure samples with their time-matched controls. The cutoff values of fold changes (FC) ≥ 1.5 or ≤ −1.5 (absolute FC ≥ 1.5) and *p*-value < 0.05 were used [26]. Additionally, false discovery rate (FDR) was applied according to the Benjamini–Hochberg method with a threshold q-value < 0.05 [26].

### 2.9. Pathway Analysis

To identify biological processes and pathways associated with the identified DEGs, an over-representation analysis (ORA) was performed. Each timepoint and dose was analyzed with ConsesusPathway DB (CPDB, release 34, accessed on 1 January2021) [27,28]. The *p*-value was calculated for each annotation set, and within each set a correction for multiple testing was performed (q-value). Cutoff values for the ORA were set at a minimum overlap of the genes with the input list of 2 and *p*-value < 0.01 for each pathway. All available mouse databases from CPDB were used.

### 2.10. STEM Analysis

The short time-series express miner (STEM) analysis was performed with the STEM tool developed by Ernst and Bar-Joseph [29]. STEM is an algorithm that compares short time-series gene expression data and clusters gene expression patterns over time, which helps to visualize and analyze microarray data in regard to the directionality of genes [29]. The analysis was performed with the log2FC expression values of all genes processed by LIMMA analysis and was grouped per dosage (1, 2, and 5 mg/kg_bw_/day of E171) over the 4 timepoints (2, 7, 14, and 21 days). The gene annotation source was set on Mus musculus. No additional cross-references, no gene location, and no normalization settings were used. The STEM clustering method was set on a maximum of 50 model profiles per analysis. Each expression of a gene was compared to the previous timepoint, and the maximum unit change in the model profiles between the timepoints was set to 2. Significance (*p*-value < 0.05) was calculated by comparing the actual number of genes per cluster to the expected number of genes per cluster by using Bonferroni correction [29]. The clustered genes that were assigned to statistically significant profiles were grouped by biological function (color) and combined into a single graph. The genes in these obtained graphs/clusters were analyzed with ORA via CPDB, as described in the previous section.

### 2.11. Network Analysis

Enrichment and network analysis were performed using the web-based online tool Metascape (20210801, updated 20 July 2021). Metascape provides gene annotation, membership analyses, and multi-gene list meta-analyses, which are based on well-adapted hypergeometric tests and Benjamini–Hochberg *p*-value correction algorithms to classify ontological parameters that contain a substantially larger set of genes common to an input list than expected [30]. Pathway enrichment analysis accessed Gene Ontology, KEGG, Reactome, and MSigDB. Based on a Kappa test score, pairwise similarities between any two enriched terms were computed. These similarity matrices were clustered by hierarchy and a 0.3 threshold was applied to create clusters. Metascape chose the most significant (lowest *p*-value, Bonferroni correction <0.05) terms within each cluster to represent the cluster in a heatmap [30]. Interactome/network analysis utilized protein–protein interactions which were captured in BioGrid, with the additional integration of InWEB_IM and OmniPath. Each network complex was analyzed via function enrichment analysis and the top three enriched terms were annotated as biological associations. All network visualizations were generated by Cytoscape [30]. Network analysis was performed with all 5% FDR-corrected DEGs after the exposure to 1 mg/kg_bw_/day of E171 for 2, 7, 14, and 21 days of exposure, with the standard settings of the express analysis function and “input as species” and “analysis as species” set to Mus musculus.

## 3. Results

### 3.1. Particle Characterization

Food-grade titanium dioxide (E171) was analyzed using quantitative TEM analysis and DLS measurements to determine Z-average and zeta potential. TEM analysis showed that E171 was comprised of two size fractions: ~64% nanoparticles (<100 nm) and ~36% microparticles (>100 nm). The median particle size (Fmin—short axis; Fmax—long axis) is displayed in Table 1. TEM pictures showed that the E171 particles tended to agglomerate (Figure 1). Similar effects were observed in the DLS analysis. E171 formed larger clusters, which increased in Z-average with increasing particle concentration. The Zeta-potential of the three different particle concentrations showed a slight decrease with increasing particle concentration, indicating a small reduction in the stability of the particle dispersion.

### 3.2. Pilot Study: Histopathology

In the pilot study, CAC^Tg/Tg^;APC^580S/+^ mice were exposed to 5 mg/kg_bw_/day of E171 or sterile water via drinking water for four weeks. E171 intake ranged from 3.5 to 5.5 mg/kg_bw_/day. No E171 particles were observed in the colon specimens. The two control mice harbored 6 and 15 tumors in the colon, with an average volume of 15.3 mm^3^ and 11.9 mm^3^, respectively. The mice exposed to E171 harbored 1, 10, and 9 tumors in their colon, with an average volume of 3.32, 48.4, and 99.1 mm^3^, respectively (Appendix A). Histological analysis of these tumors in the control and E171-treated mice (Figure 2) showed well-differentiated adenocarcinomas. Mice exposed to E171 showed adenocarcinomas with enhanced hyperplasia in epithelial cells as well as epithelial cell infiltration in the muscle layer.

### 3.3. Phase I: Tumor Formation Study

The CAC^Tg/Tg^;APC^580S/+^ Tg mouse model was used to study tumor formation in the colon after exposure to 5 mg/kg_bw_/day of E171 by intragastric administration for 9 weeks. Our experiment showed no effect on the bodyweight of these Tg mice, following the exposure to E171 (Figure 3). Similarly, no significant effect on organ weight of the colon, liver, or spleen was detected (Figure 3). However, Figure 4A shows a tendency that the average number of tumors per mouse increased after 7 and 9 weeks of exposure. Additionally, a statistically nonsignificant increase in the number of mice with tumors was registered after E171 exposure, when compared to the controls (Figure 4B).

### 3.4. Phase II: Gene Expression Study

To study gene expression, CAC^Tg/Tg^;APC^580S/+^ Tg mice were exposed to 1, 2, and 5 mg/kg_bw_/day of E171 for 2, 7, 14, and 21 days. One tumor was found in the colon of one mouse after exposure to 1 mg/kg_bw_/day of E171 for 21 days (data not shown) and was registered. No other tumors at any other timepoint or concentration were detected. DEGs were identified with LIMMA and corrected for multiple testing (FDR 5%). Table 2 shows the complete output of the LIMMA analysis including the number of up- and down-regulated genes at different timepoints and doses. No clear dose–response curve was observed over time while analyzing the number of DEGs (Figure 5), except for a time-dependent increase in DEGs following the exposure to 1 mg/kg_bw_/day of E171. The number of DEGs following exposure to 2 mg/kg_bw_/day was continuously the lowest, compared with the other two dosages. The majority of 5% FDR-corrected DEGs was detected following the exposure of 1 mg/kg_bw_/day of E171. All further analyses focused only on the dosage of 1 mg/kg_bw_/day of E171. Figure 6 shows the Venn diagram of 5% FDR-corrected DEGs (absolute FC ≥ 1.5 and q-value < 0.05) for mice exposed to 1 mg/kg_bw_/day of E171 per timepoint and their overlap. A total number of three genes were differentially expressed throughout all four timepoints. These three genes were of the D site albumin-promoter-binding protein (Dbp) and nuclear receptor subfamily 1, group D, member 1/2 (Nr1d1, Nr1d2). After 2, 7, and 14 days of E171 exposure, the genes Aryl hydrocarbon receptor nuclear translocator-like (Arntl/Bmal1) and nuclear-factor-regulated interleukin 3 (Nfil3) were modulated. Nuclear factor of kappa light polypeptide gene enhancer in B-cell inhibitors, zeta (Nfkbiz) was modulated at 2, 14, and 21 days after exposure to E171. After 7, 14, and 21 days of E171 exposure, the gene RAR-related orphan receptor gamma (Rorc) was modulated. Ring finger protein 125 (Rnf125) and protein yippee-like 2 (Ypel2) were modulated at 2 and 14 days, while neuronal PAS domain protein 2 (Npas2), apolipoprotein L7a (Apol7a), and tensin 4 (Tns4) were modulated at 2 and 7 days following E171 exposure. Figure 7 shows a heatmap of all significantly expressed genes (absolute FC ≥ 1.5, q-value < 0.05) that were modulated at one or more timepoints, following the exposure to 1 mg/kg_bw_/day of E171.

### 3.5. Pathway Analysis

The pathway analysis was performed with the corresponding DEGs that passed absolute FC ≥ 1.5 and q-value < 0.05 (FDR 5%) for each timepoint and dose. The resulting pathways are shown in Table 3. Only DEGs following an exposure to 1 mg/kg_bw_/day of E171 resulted in the detection of significantly altered genes and their consequent involvement in genetic pathways. Pathways related to cancer, cell cycle, circadian rhythm, metabolism, post-translational modification, and gene expression (transcription) were identified. The pathways of the circadian rhythm, exercise-induced circadian rhythm, nuclear receptor, and nuclear receptor transcription were modulated at all timepoints. On day 2, additional pathways relating to the cell cycle, disease, cancer, metabolism, and post-translational modification were identified. On day 21, additional pathways related to gene expression were detected. Appendix A show the ORA pathway analysis of DEGs (absolute FC ≥ 1.5, *p*-value < 0.05) without FDR correction after exposure to 1, 2, and 5 mg/kg_bw_/day of E171. The ORA analysis via CPDB showed that on day 2 of exposure to 1 mg/kg_bw_/day of E171 pathways related to cell cycle and cell proliferation were modulated. More specifically, ERBB-2-, ERBB-4-, and PI3K-related pathways, as well as pathways associated with the circadian rhythm, post-translational modification, and nuclear receptor (transcription)-related pathways were identified. After 7 days of exposure to 1 mg/kg_bw_/day of E171, pathways related to cholesterol and lipid metabolism, modulation of the cell cycle, and signaling (e.g., p53 signaling) were found. Exposure to 1 and 5 mg/kg_bw_/day of E171 for 14 days showed colorectal-cancer-related gene modulation in pathways associated with cell cycle and proliferation (PLK1, TP53 pathways), as well as DNA damage response. After 21 days of exposure to 1 and 5 mg/kg_bw_/day of E171, our analysis revealed genes modulated in pathways that are associated with lipid and cholesterol metabolism as well as cell proliferation (FGFR1). This analysis of non-FDR-corrected DEGs showed additional modulations of genes involved in pathways that are strongly associated with the development of CRC and the general development and progression of tumors in the intestinal tract.

### 3.6. STEM Analysis

Figure 8 shows the output of the STEM analysis for all genes that were identified as significant. Genes are grouped by dosage and over time and were combined into clusters according to their biological function. The pathways associated with each of the clusters and dosage can be found in Table 4. Following the exposure to 1 mg/kg_bw_/day of E171, cluster 1 showed an upregulation of pathways related to G alpha signaling events and olfactory signaling. Genes assigned to cluster 2 were associated with the pathways of olfactory transduction/signaling and neuroactive ligand–receptor interactions. Cluster 3 showed the downregulation of genes involved in pathways related to circadian rhythm, cytokine–cytokine receptor interactions, O-glycosylation, and RAF/MAPK signaling cascade. Following the exposure to 2 mg/kg_bw_/day of E171, two clusters were significantly altered, which correlates with the overall lower number of DEGs that were observed at this timepoint. Cluster 1 showed an increased expression of genes involved in pathways related to immune responses, including cell adhesion molecules (CAM), complement cascade, hematopoietic cell lineage, immunoregulatory interactions, and B-cell receptor signaling. The genes present in cluster 2 did not correlate with any known pathways. Tg mice exposed to 5 mg/kg_bw_/day of E171 showed the highest number of genes and pathways that were significantly modulated. Genes combined in cluster 1 were related to signaling, inflammation, immunoregulatory responses, and disease. Cluster 2 showed modulation of genes involved in the cell cycle (e.g., TP53- and APC-related pathways), tumor suppression, and signaling. Cluster 3 showed an upregulation of genes related to olfactory and GPCR signaling, as well as extracellular matrix organization. The genes combined in cluster 4 showed modulation of an inflammatory mediator regulation of transient receptor potential (TRP) channel. Overall, STEM analysis showed the temporal development of genes related to pathways involved in signaling, circadian rhythm, inflammation/immune responses, and cell cycle and tumor development (Table 4). It highlighted the significance of pathways identified via LIMMA/ORA, such as circadian rhythm, glycosylation, nuclear receptor signaling, inflammation, and cell-cycle-related events.

### 3.7. Functional Enrichment Analysis and Network Analysis

Metsacape heatmap enrichment analysis identified similar alterations of genetic pathways as previously described by ORA and STEM analysis. Figure 9 shows the enrichment heatmap of the significant (*p*-value < 0.05) pathways that were identified when analyzing all DEGs (absolute FC ≥ 1.5 and q-value < 0.05) after the exposure of Tg mice to 1 mg/kg_bw_/day of E171 for 2, 7, 14, and 21 days. These pathways included rhythmic processes, hormone-mediated signaling pathways, nuclear receptor transcription pathways, retinol metabolism, negative regulation of inflammatory response, platelet-derived growth factor receptor signaling pathways, small cell lung cancer, muscle tissue development, O-linked glycosylation, and regulation of T-cell differentiation. Figure 10 shows the functional network relationships between the identified biological processes and their interactions, indicating the connection between genes and the pathways they are involved in through nodes.

## 4. Discussion

In this study, we examined the effects of food-grade E171 exposure in a CAC^Tg/Tg^;APC^580S/+^ transgenic mouse model. The pilot study showed histopathological changes in the colon of this Tg mouse model, including hyperplasia around the epithelial cells and the invasion of epithelial cells into the muscle layer. In phase I of this experimental setup, exposure to E171 increased the average number of colonic tumors per mouse, as well as the number of mice bearing tumors, compared with the controls, however without statistical significance. Furthermore, this study showed transcriptome changes in the distal colon that indicated genes involved in genetic pathways that may contribute to the onset of CRC within the same model. This study supports the observed effects of E171 on the development of colorectal tumors, as previously reported in a chemically induced CRC murine mouse model (AOM/DSS); although, the effects of E171 in the Tg mouse model were less pronounced [12,15].

Histological examination of colonic specimens from the pilot study revealed an increase in tumor size and enhanced hyperplasia at the bottom of the adenocarcinomas. We observed large hyperplastic areas, showing loss of tissue architecture, nuclear enlargement, and increased nucleus-to-cytoplasm ratio (anaplasia), but also infiltrated epithelial cells in the muscle layer, which denotes malignancy. Although infiltrated epithelial cells were observed in both, control and treated mice, the amount of cell infiltration was higher in E171 exposed mice [31].

The tumor formation study indicates an earlier onset of tumor formation and differences in tumor size, following the exposure to E171. The occurrence of these changes in tissue architectures and tumors size might be explained by the findings of our gene expression study. The gene expression experiment in this Tg mouse model showed that the exposure to 1, 2, and 5 mg/kg_bw_/day of E171 did not result in a clear dose–time response in the expression changes, except for a tendency of a time-dependent increase in the number of DEGs following the exposure to 1 mg/kg_bw_/day. The absence of a linear dose–response curve over time might be a consequence of the aggregation and agglomeration of the TiO_2_ particles at higher concentrations, as observed by us previously [32]. Based on earlier studies, we hypothesized that E171 exposure enhances colorectal cancer development by modulation of gene expression changes in signal transduction, oxidative stress, inflammation, DNA damage/repair, and interferences with the immune system [14,15].

This Tg mouse model revealed consistent modulations of genes in pathways related to the circadian rhythm, namely the “circadian rhythm” and the “exercise-induced circadian rhythm”, following ORA via CPDB. The modulation of genes in these pathways is linked to CRC through their involvement in transcriptional and translational networks and nuclear signaling [33]. Our study revealed the modulation of several genes in these pathways, including *Per3*, *Cry1*, *Nr1d1*, *Nr1d2*, *Arntl* (*Bmal1*), *Npas2*, and *Rorc*. Alterations of genes in the *Npas* and *Per* families were associated with a dysfunctional cell cycle, resulting in higher susceptibility for DNA-damage-induced cancer development and overall survival in CRC patients [34,35,36]. Epidemiological studies provide an additional link and state that the alteration of the circadian clock in shift workers has been identified as a probable carcinogen to humans [37]. Cell cycle, cell signaling, and proliferation are also rhythmically expressed and hence partially regulated by the expression of circadian clock genes e.g., the *Per* and *Cry* families, making them a potential target for modulation of circadian-rhythm-related pathways [36,38]. Furthermore, a comparison of mRNA expression levels of tumor tissue and non-tumor mucosa of human colorectal specimens showed a decrease in *Cry1* and *Cry2* [39], whereas a different study reported *Cry1* overexpression in CRC cell lines and specimens, suggesting an association with a poor prognosis in patients with CRC [38].

Another potential driver in the development of CRC is the redox-oscillating *Arntl* (*Bmal1*) gene [40]. It has been shown to inhibit tumor proliferation through G2-M phase cell cycle arrest and is known as an important regulator of the p53/p21 tumor suppressor pathway [39,41,42]. Downregulation of *Arntl* (*Bmal1*) reduces the cells’ ability to induce cell cycle arrest upon p53 activation in response to cellular stress signals or DNA damage [39]. The formation of heterodimers between *Npas2* and *Arntl* (*Bmal1*) may regulate the transcription of tumor cell growth and survival, further indicating a potential tumor-suppressing effect associated with these genes [36,42]. *Arntl* (*Bmal1*) has been further associated with the control of extracellular matrix organization, through dysregulation of matrix metalloproteinase 2 and 9 (MMP-2, -9) [43,44]. MMPs, especially the gelatinases MMP-2/-9, are strongly related to the development of metastasis and secondary tumors in CRC [44]. Our STEM analysis revealed modulation of extracellular matrix degradation, including genes that modulate pathways related to activation of matrix metalloproteinase.

Additional modulation of genes in pathways following the oral exposure to 1 mg/kg_bw_/day of E171 for 2 days were linked to glycosylation, O-linked glycosylation, and O-linked glycosylation of TSR-containing proteins. The genes *Adamts4*, *Adamts14*, and *Spon2* were significantly modulated in these pathways, and further indicate a correlation between E171 exposure and extracellular matrix organization related to the development of CRC. The *Adamts* gene family encodes for proteinases that are responsible for extracellular matrix degradation and regulation. *Adamts4* has been shown to contribute to the development and progression of CRC, especially due to its effects on tumor growth, the regulation of macrophages, and the influence on the inflammatory microenvironment in cancer [45]. A correlation between *Adamts4* and cancer progression has also been published by Filou et al. (2015), which indicates the involvement of these collagen-processing proteases in CRC [46]. Overexpression of *Spon2* is highly associated with colon cancer metastasis, which displays one of the most feared side effects of CRC [47]. Contrary to these findings, this Tg mouse model showed downregulation of *Spon2* expression after 2 days of exposure to 1 mg/kg_bw_/day of E171, which might result from the short exposure time and potential compensation mechanisms of the organism. The regulation of the extracellular matrix environment and its involvement in the expression of proteins, growth factors, chemokines, and cell adhesion molecules overall contribute to the high risk of metastasis within CRC patients and showcase an important hallmark of cancer [44].

Genes relating to pathways of nuclear receptor signaling were consistently modulated after exposure to 1 mg/kg_bw_/day of E171. Nuclear receptors and their function as a sensor for dietary or endogenous stimuli are responsible for the translation of nutritional or hormonal signals into transcriptional modifications [48]. These are commonly regulated by hormones and metabolites of steroid retinoids (vitamin A metabolites), vitamin D, fatty acids, bile acids, and other dietary derived lipids [48]. Our study demonstrated the modulation of nuclear receptors over time, involving the genes *Rarb* (day 2), *Nr1d1*/*Nr1d2* (day 2, 7, 14, and 21), and *Rorc* (day 7, 14, and 21). *Rarb* (*Rar-ß*) plays a critical role in the progression of several human cancers, including CRC, where it is responsible for the transcription of genes involved in cellular differentiation and acts as a potential tumor suppressor, through the subsequent modulation of the retinoic acid response element (RARE) [49,50]. Furthermore, *Rarb* and *Rorc* are involved in the regulation of ß-catenin/WNT signaling [51]. While a direct modulation of transcripts relating to ß-catenin/WNT signaling was not found in this Tg mouse mode, many studies suggest a link between the dysregulation of this pathway and CRC [52,53]. *Rorc* is dysregulated in a multitude of cancers and is likely to participate in carcinogenesis through the modulation of IL-17, androgen receptors, and protein arginine N-methyltransferase 2 (*Prmt2*), leading to a ligand-dependent interaction and co-regulation of mechanisms associated with the development of inflammatory diseases, homeostasis, and circadian rhythm [54,55]. Furthermore, *Rorc* functions as a transcription factor for *RORyt*, which is involved in the maturation of thymocytes and T-helper cells (Th), particularly Th17, which main function is the production of IL-17 [56]. Modulation of Th1/Th17 has been shown by Bettini et al. (2017), in rats exposed to 10 mg/kg_bw_/day of E171 [13]. Changes in the regulation of T cell differentiation have also been identified in our functional enrichment analysis. *Nr1d1*/*Nr1d2* are additional nuclear receptor-associated core-clock genes that are modulated after exposure to E171. *Nr1d1* especially impacts the circadian rhythm phenotype of *Myc*, *Wee1*, and *Tp53*; hence, it is involved in the processes of cell proliferation, apoptosis, and cell migration [57]. Another potential tumor suppressor gene is the *Dhrs3* gene, which together with *Rarb* was modulated in pathways relating to retinol metabolism. *Dhrs3* and *Rarb* have been correlated with various types of cancer, including CRC and gastric cancer. *Dhrs3* as a tumor suppressor gene has been suggested to play a critical role in connecting the retinol metabolism with the circadian rhythm, leading to an alteration of immune responses and cellular metabolism [58,59,60].

The cell-cycle-dependent checkpoint-altered gene expression of *Skp2*, *Mcm6*, and *E2f2* were identified. *Skp2* modulation, and the loss of its substrates *p27* and *Mcm6*, and the *E2f* family are clinical markers for a poor outcome in CRC, its malignancy, and CRC tumor growth [61,62,63].

Additionally, identified pathways (Appendix A Appendix A; absolute FC ≥ 1.5, *p*-value < 0.05) included PI3K, ERBB2/HER2, ERBB-4, PLK1, TP53, and FGFR-related pathways, further highlighting the involvement of tumor suppressor genes and cancer-related pathways in CRC development, following the exposure to the food additive E171 [64,65]. These pathways correlated strongly with events detected by STEM analysis, particularly the TP53-, PI3K-, and RAF/MAPK-related pathways.

Previously, we studied gene expression responses after exposure to 5 mg/kg_bw_/day of E171 via intragastric gavage in various mouse models, including normal BALB/c mice and colitis-induced AOM/DSS mice [14,15]. Across these three different models, similarities were found in the modulation of pathways related to signal transduction, cell cycle events, metabolism, inflammation, and tumor development. More specifically, we found the modulation of olfactory/GPCR-related pathways, inflammation-related pathways, modulation of extracellular spaces, activation of the immune system, metabolic changes, DNA damage/repair, and cancer-related signaling pathways (e.g., MAPK, PI3K) to be in common in this study [14,15].

## 5. Conclusions

In this comprehensive study, using a *CAC^Tg/Tg^; APC^580S/+^* transgenic mouse model for colorectal cancer, we showed increased tumor growth and progression, but found no statistically significant increase in tumor formation induced by E171. We used ORA and STEM analyses to identify significantly modulated genes in genetic pathways and their temporality, to define mechanisms that are potentially involved in the increased tumor formation observed after exposure to E171. Our data suggest that E171 may contribute to tumor formation and progression by modulation of pathways related to inflammation, activation of immune responses, cell cycle events, and cancer signaling. These findings can be of relevance in the ongoing debate on the safety evaluation of E171 and contribute to the identification of molecular mechanisms related to E171-induced genotoxicity and carcinogenicity. Further toxicity studies are needed to evaluate the safety of E171 and other metal-based nanomaterials, which are used as food additives or food packaging materials.

## Figures and Tables

**Figure 1 nanomaterials-12-01256-f001:**
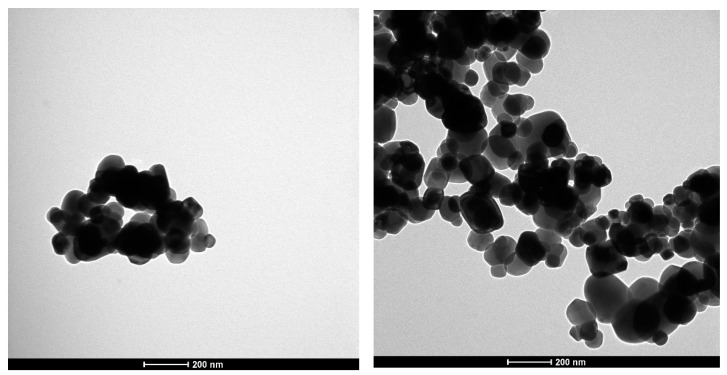
Exemplary transmission electron microscopy (TEM) images of food grade E 171 in sterile water (60,000× magnification).

**Figure 2 nanomaterials-12-01256-f002:**
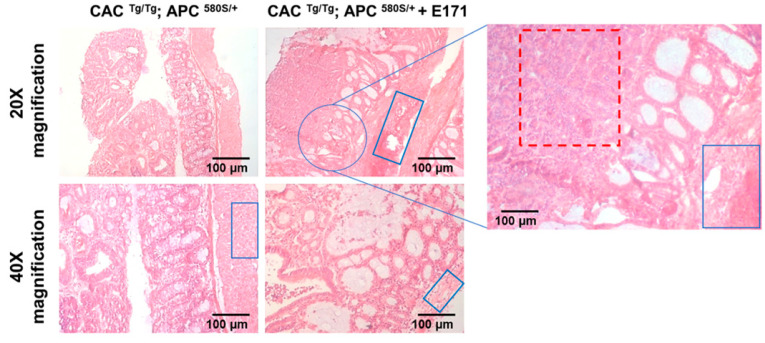
Histopathological analysis of tumors in control and E171-treated mice showed well-differentiated adenocarcinomas. The mice exposed to E171 additionally showed enhanced hyperplasia in the epithelial cells as well as epithelial cell infiltration in the muscle layer of the adenocarcinomas. Red squares—epithelial cell hyperplasia and anaplasia; blue squares—epithelial cell infiltration into the muscle layer (hyperplasia).

**Figure 3 nanomaterials-12-01256-f003:**
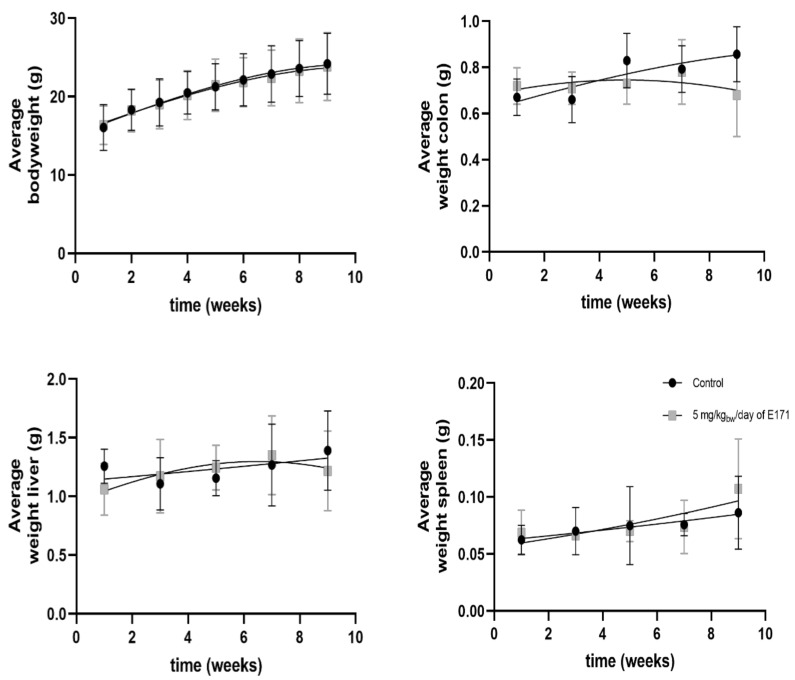
Average bodyweight and organ weight of Tg mice following intragastric exposure to 5 mg/kg_bw_/day of E171 over time (n = 78). One mouse was euthanized ahead of the schedule due to rectal prolapse. A second mouse from the control group was ruled as an outlier due to severe tumor formation. These mice were not included in the graphs. After euthanasia, colon, liver, and spleen were weighed. The black data series corresponds to the control mice exposed to sterile water and the grey data series is attributed to the mice exposed to 5 mg/kg_bw_/day of E171 via intragastric administration. Data is presented as the mean +/− standard deviation of each timepoint.

**Figure 4 nanomaterials-12-01256-f004:**
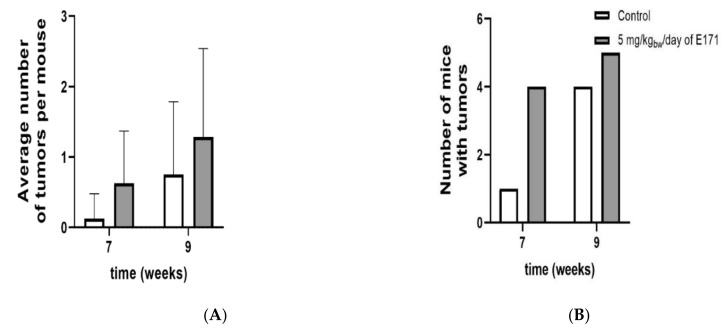
Tumor formation in the colon of Tg mice exposed to 5 mg/kg_bw_/day of E171 via intragastric gavage for 7 and 9 weeks. (**A**) shows the average number of tumors per mouse. (**B**) shows the number of mice bearing tumors. Data in (**A**) is presented as the mean of each group (n = 8 for each group for 7 and 9 weeks; n = 7 for control 7 weeks and 5 mg/kg_bw_/day of E171 for 9 weeks). E171 exposure showed a statistically nonsignificant increase in the number of tumors per mouse as well as the number of mice with tumors.

**Figure 5 nanomaterials-12-01256-f005:**
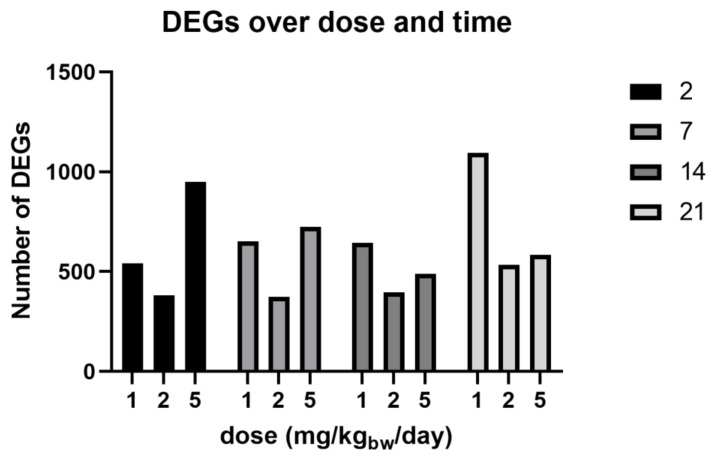
Number of differentially expressed genes (DEGs) after exposure to 1, 2, and 5 mg/kg_bw_/day of E171 in the colon of Tg mice. Bars correspond with an absolute FC ≥ 1.5. The legend indicates the days of exposure. Exposure to 1 mg/kg_bw_/day of E171 showed a time-dependent increase in DEGs from 2 to 21 days. A measure of 2 mg/kg_bw_/day of E171 continuously showed the lowest response throughout all four timepoints, while 5 mg/kg_bw_/day of E171 resulted in a decreasing number of DEGs with an increased exposure time. The exact number of DEGs per timepoint and dose can be found in Table 2.

**Figure 6 nanomaterials-12-01256-f006:**
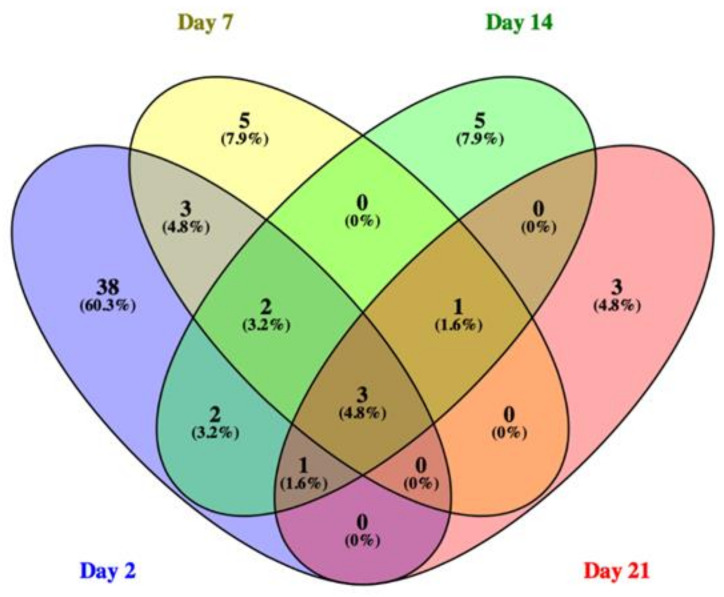
The Venn diagram shows the number of 5% FDR-corrected DEGs (absolute FC ≥ 1.5 and q-value < 0.05) and their overlap after the exposure to 1 mg/kg_bw_/day of E171 at 2, 7, 14, and 21 days. The three genes, Dbp, Nr1d1, and Nr1d2, were continuously modulated at all timepoints. After 2, 7, and 14 days of exposure, Arntl/Bmal1 and Nfil3 were modulated. Rorc was modulated at days 7, 14, and 21, while Nfkbiz was modulated after 2, 14, and 21 days of exposure to 1 mg/kg_bw_/day of E171.

**Figure 7 nanomaterials-12-01256-f007:**
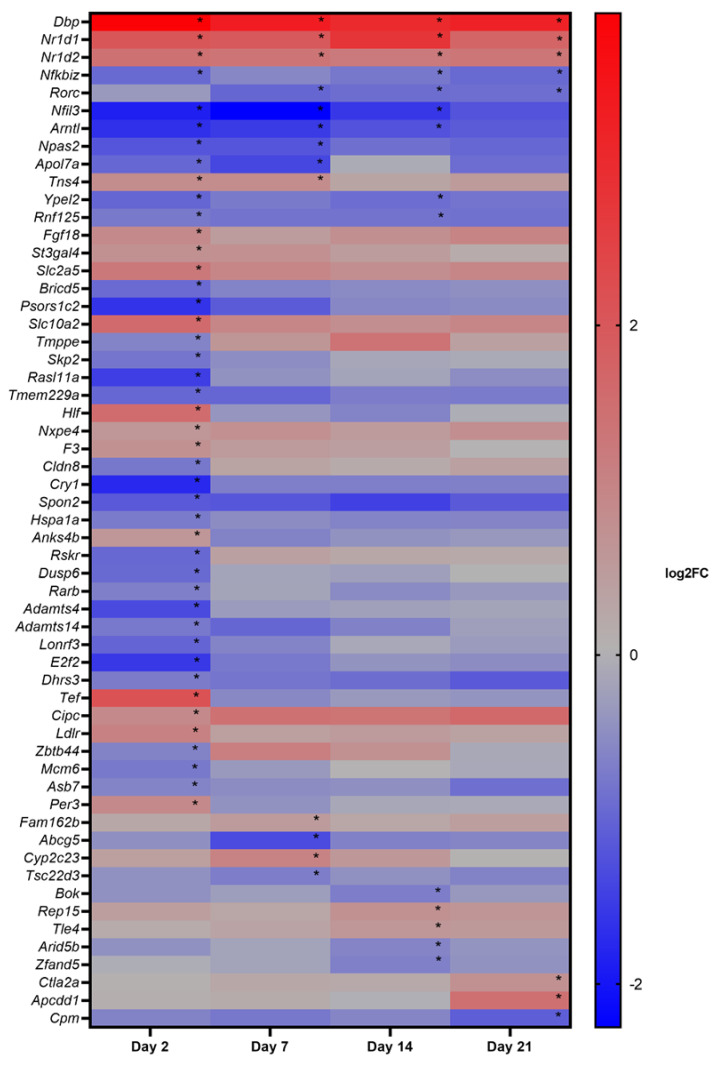
Heatmap of all genes that were differentially expressed (absolute FC ≥ 1.5, q-value < 0.05) at one or more timepoints, following the exposure to 1 mg/kg_bw_/day of E171. The rows represent the differentially expressed genes, while the columns represent the expression value for the timepoints 2, 7, 14, and 21 days. Red and blue colors indicate the log2FC: red stands for upregulation and blue for downregulation of the gene, in comparison with its time-matched control. The order of genes is based on the number of timepoints a gene was significantly differentially expressed (marked by *).

**Figure 8 nanomaterials-12-01256-f008:**
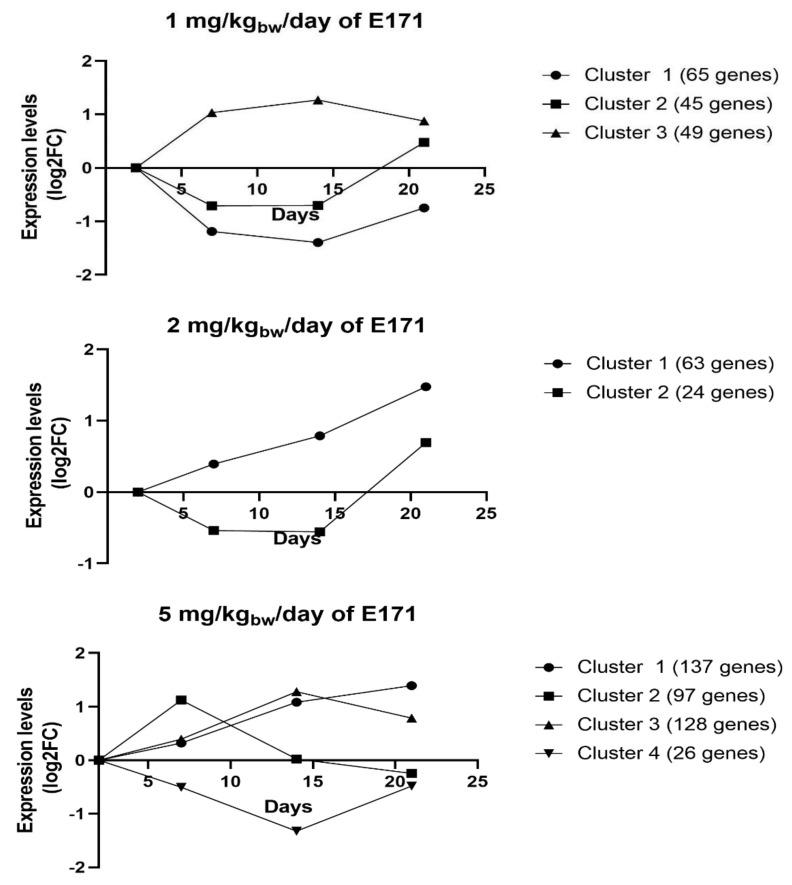
STEM analysis was performed with all genes that passed the preprocessing step to examine their temporality. All significant genes that were assigned to a profile were clustered by biological function and are represented in this figure. The *X*-axis corresponds with the timepoints 2, 7, 14, and 21 days, and the *y*-axis indicates the expression based on log2FC. Original STEM output profiles and assigned genes per profile can be found in the y data (Appendix A).

**Figure 9 nanomaterials-12-01256-f009:**
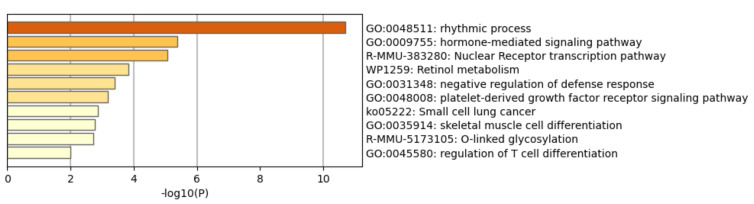
Metascape functional enrichment heatmap following the analysis of all 5% FDR-corrected DEGs (absolute FC ≥ 1.5, q-value < 0.05) after the exposure to 1 mg/kg_bw_/day of E171 in a Tg mouse model. Bar graphs of enriched terms across the input gene list are colored by *p*-values. Significantly altered genetic pathways included rhythmic processes, signaling, nuclear transcription pathway, retinol metabolism, negative regulation of defense responses, platelet-derived growth factor signaling, small cell lung cancer, skeletal muscle cell differentiation, O-linked glycosylation, and regulation of T-cell differentiation, confirming genetic alteration in pathways that were identified by ORA and STEM analysis.

**Figure 10 nanomaterials-12-01256-f010:**
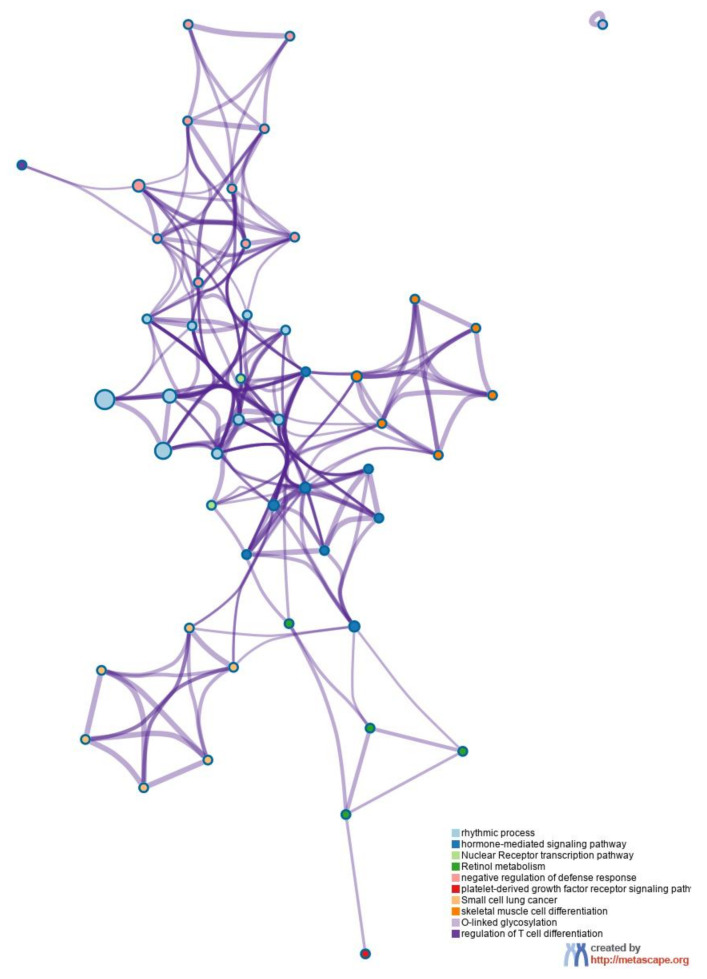
Network showing the interconnection of enriched terms. Clustered genes were typically close to each other and colored the same way when belonging to the same biological process, indicated by the legend in the bottom right corner. Edges linked similar terms, where thicker edges indicate higher similarity. The functional network shows genes according to their function and interaction. All 5% FDR-corrected DEGs (absolute FC ≥ 1.5, q-value < 0.05) following the exposure to 1 mg/kg_bw_/day of E171 from all timepoints were used to construct the network via Metascape.

**Table 1 nanomaterials-12-01256-t001:** Summary of particle characterization of food-grade E171 dispersed in sterile water. Particle characterization included determination of median particle size, percentage of particle >100 nm obtained from the quantitative TEM analysis, Z-average PDI, and zeta-potential obtained from DLS measurements.

	TEM			DLS		Zeta-Potential
	Fmax (nm) Median ± 95% CI	Fmin (nm) Median ± 95% CI	Particles < 100 nm (%)	Z-Average (nm) Mean ± SD	PDI Mean ± SD	(mV) Mean ± SD
1 mg/mL	107 (±8.3)	79 (±5.4)	~64	315.3 (±92.4)	0.246 (±0.02)	−29.9 (±6.0)
2 mg/mL	106 (±11.4)	75 (±7.2)	~64	318.1 (±96.6)	0.218 (±0.02)	−28.7 (±7.8)
5 mg/mL	110 (±7.7)	79 (±4.8)	~63	348.7 (±140.5)	0.192 (±0.01)	−27.1 (±9.4)

CI—confidence interval; SD—standard deviation.

**Table 2 nanomaterials-12-01256-t002:** DEGs after LIMMA analysis of the microarray data obtained from Tg mice exposed to E171 at 1, 2, and 5 mg/kg_bw_/day, including absolute FC, the number of up- and downregulated genes, *p*-value, and q-value, as well as a combination of absolute FC and *p*/q-values. The DEGs in bold (absolute FC ≥ 1.5 and q-value < 0.05) were used for ORA pathway analysis.

	Day 2			Day 7			Day 14			Day 21		
Dose (mg/kg_bw_/day)	**1**	**2**	**5**	**1**	**2**	**5**	**1**	**2**	**5**	**1**	**2**	**5**
absolute FC ≥ 1.5	542	382	950	650	373	724	643	396	489	1095	534	584
Upregulated	231	124	262	360	202	341	202	188	168	499	352	386
Downregulated	311	258	688	290	171	383	441	208	321	596	182	198
*p*-value < 0.05	1657	752	1534	1403	988	1025	1421	501	1180	1517	562	1172
q-value < 0.05	61	0	1	15	1	0	16	2	7	8	0	0
absolute FC ≥ 1.5 and *p*-value < 0.05	311	129	337	328	156	357	282	135	238	489	129	246
absolute FC ≥ 1.5 and q-value < 0.05	**49**	0	1	**14**	1	0	**14**	1	6	**8**	0	0

**Table 3 nanomaterials-12-01256-t003:** Pathway over-representation analysis (ORA) of the 5% FDR-corrected DEGs following the exposure to 1 mg/kg_bw_/day of E171 (absolute FC ≥ 1.5, q-value < 0.05). Pathways were grouped by timepoint and biological function.

Biological Function	Day	Pathway	Source	*p*-Value	q-Value
Cancer	2	Small cell lung cancer–Mus musculus (mouse)	KEGG	2.16 × 10^−3^	1.73 × 10^−2^
	2	Gastric cancer–Mus musculus (mouse)	KEGG	6.63 × 10^−3^	2.84 × 10^−2^
Cell cycle	2	Cell cycle	Wikipathways	1.83 × 10^−3^	1.73 × 10^−2^
	2	Mitotic G1-G1/S phases	Reactome	2.16 × 10^−3^	1.73 × 10^−2^
	2	Orc1 removal from chromatin	Reactome	2.43 × 10^−3^	1.73 × 10^−2^
	2	Cell cycle–Mus musculus (mouse)	KEGG	4.96 × 10^−3^	2.64 × 10^−2^
	2	Cyclin D associated events in G1	Reactome	6.77 × 10^−3^	2.84 × 10^−2^
	2	G1 Phase	Reactome	6.77 × 10^−3^	2.84 × 10^−2^
	2	Switching of origins to a post-replicative state	Reactome	7.09 × 10^−3^	2.84 × 10^−2^
Circadian rhythm	2	Circadian rhythm–Mus musculus (mouse)	KEGG	1.48 × 10^−8^	9.49 × 10^−7^
	2	Exercise-induced circadian regulation	Wikipathways	1.10 × 10^−5^	3.53 × 10^−4^
Disease	2	Hepatitis C–Mus musculus (mouse)	KEGG	8.93 × 10^−3^	3.17 × 10^−2^
Gene expression (transcription)	2	Nuclear receptor transcription pathway	Reactome	3.25 × 10^−4^	4.16 × 10^−3^
	2	Nuclear receptors	Wikipathways	4.74 × 10^−3^	2.64 × 10^−2^
Metabolism	2	Retinol metabolism	Wikipathways	3.73 × 10^−3^	2.39 × 10^−2^
	2	Signaling by retinoic acid	Reactome	7.74 × 10^−3^	2.92 × 10^−2^
Post-translational modification/ metabolism	2	O-glycosylation of TSR-domain-containing proteins	Reactome	1.20 × 10^−4^	2.12 × 10^−3^
	2	O-linked glycosylation	Reactome	1.33 × 10^−4^	2.12 × 10^−3^
Circadian rhythm	7	Circadian rhythm–Mus musculus (mouse)	KEGG	1.11 × 10^−8^	8.87 × 10^−8^
	7	Exercise-induced circadian regulation	Wikipathways	1.29 × 10^−5^	3.45 × 10^−5^
Gene expression (transcription)	7	Nuclear receptor transcription pathway	Reactome	1.14 × 10^−5^	3.45 × 10^−5^
	7	Nuclear receptors	Wikipathways	5.48 × 10^−4^	1.10 × 10^−3^
Circadian rhythm	14	Circadian rhythm–Mus musculus (mouse)	KEGG	2.48 × 10^−6^	1.98 × 10^−5^
	14	Exercise-induced circadian regulation	Wikipathways	1.29 × 10^−5^	3.45 × 10^−5^
Gene expression (transcription)	14	Nuclear receptor transcription pathway	Reactome	1.14 × 10^−5^	3.45 × 10^−5^
	14	Nuclear receptors	Wikipathways	5.48 × 10^−4^	1.10 × 10^−3^
Circadian rhythm	21	Circadian rhythm–Mus musculus (mouse)	KEGG	7.84 × 10^−5^	2.74 × 10^−4^
	21	Exercise-induced circadian regulation	Wikipathways	2.33 × 10^−4^	4.08 × 10^−4^
Gene expression (transcription)	21	Nuclear receptor transcription pathway	Reactome	9.63 × 10^−7^	6.74 × 10^−6^
	21	Nuclear receptors	Wikipathways	1.23 × 10^−4^	2.87 × 10^−4^
	21	Generic transcription pathway	Reactome	3.31 × 10^−3^	4.64 × 10^−3^
	21	RNA polymerase II transcription	Reactome	5.19 × 10^−3^	6.06 × 10^−3^
	21	Gene expression (transcription)	Reactome	7.44 × 10^−3^	7.44 × 10^−3^

**Table 4 nanomaterials-12-01256-t004:** Pathways over-representation analysis (ORA) resulting from STEM analysis of the gene expression changes from all genes following the exposure to 1, 2, and 5 mg/kg_bw_/day of E171 over time. Profile colors indicate gene expression profiles that were assigned to a similar biological process.

Biological Function	Dose (mg/kg_bw_/Day)	Cluster	Pathway	Source	*p*-Value	q-Value
Signaling	1	1	G alpha (s) signaling events	Reactome	1.77 × 10^−3^	1.40 × 10^−2^
	1	1	Olfactory transduction–Mus musculus (mouse)	KEGG	2.33 × 10^−3^	1.40 × 10^−2^
	1	1	Olfactory signaling pathway	Reactome	4.45 × 10^−3^	1.78 × 10^−2^
Signaling	1	2	Olfactory transduction–Mus musculus (mouse)	KEGG	3.59 × 10^−7^	4.67 × 10^−6^
	1	2	Neuroactive ligand–receptor interaction–Mus musculus (mouse)	KEGG	3.02 × 10^−3^	1.96 × 10^−2^
	1	2	Olfactory signaling pathway	Reactome	5.72 × 10^−3^	2.48 × 10^−2^
Signaling	1	3	RAF/MAP kinase cascade	Reactome	5.33 × 10^−3^	2.89 × 10^−2^
	1	3	MAPK1/MAPK3 signaling	Reactome	5.79 × 10^−3^	2.89 × 10^−2^
	1	3	MAPK family signaling cascades	Reactome	8.20 × 10^−3^	3.59 × 10^−2^
Circadian rhythm	1	3	Circadian rhythm–Mus musculus (mouse)	KEGG	2.53 × 10^−5^	8.86 × 10^−4^
	1	3	Exercise-induced circadian regulation	Wikipathways	4.62 × 10^−3^	2.89 × 10^−2^
Disease	1	3	Lung fibrosis	Wikipathways	4.81 × 10^−3^	2.89 × 10^−2^
Immune response	1	3	Cytokine–cytokine receptor interaction–Mus musculus (mouse)	KEGG	9.41 × 10^−3^	3.66 × 10^−2^
Post-translational modification/Metabolism	1	3	O-glycosylation of TSR domain-containing proteins	Reactome	2.30 × 10^−3^	2.89 × 10^−2^
Signaling	1	3	Hedgehog signaling pathway–Mus musculus (mouse)	KEGG	3.88 × 10^−3^	2.89 × 10^−2^
Disease	2	1	Epstein–Barr virus infection–Mus musculus (mouse)	KEGG	4.85 × 10^−3^	2.18 × 10^−2^
Immune response	2	1	Cell adhesion molecules (CAMs)–Mus musculus (mouse)	KEGG	1.46 × 10^−3^	1.39 × 10^−2^
	2	1	Regulation of complement cascade	Reactome	1.59 × 10^−3^	1.39 × 10^−2^
	2	1	Complement cascade	Reactome	2.31 × 10^−3^	1.39 × 10^−2^
	2	1	B cell receptor signaling pathway–Mus musculus (mouse)	KEGG	6.42 × 10^−3^	2.21 × 10^−2^
	2	1	Hematopoietic cell lineage–Mus musculus (mouse)	KEGG	7.36 × 10^−3^	2.21 × 10^−2^
	2	1	Immunoregulatory interactions between a lymphoid and a non-lymphoid cell	Reactome	8.76 × 10^−3^	2.25 × 10^−2^
Disease	5	1	Type I diabetes mellitus–Mus musculus (mouse)	KEGG	2.20 × 10^−3^	9.83 × 10^−3^
Immune response	5	1	Immunoregulatory interactions between a lymphoid and a non-lymphoid cell	Reactome	4.33 × 10^−7^	2.03 × 10^−5^
	5	1	Hematopoietic cell lineage–Mus musculus (mouse)	KEGG	4.47 × 10^−6^	1.40 × 10^−4^
	5	1	Immune system	Reactome	1.22 × 10^−5^	2.86 × 10^−4^
	5	1	Adaptive immune system	Reactome	4.45 × 10^−5^	5.23 × 10^−4^
	5	1	Chemokine receptors bind chemokines	Reactome	5.05 × 10^−5^	5.27 × 10^−4^
	5	1	Cell adhesion molecules (CAMs)–Mus musculus (mouse)	KEGG	1.16 × 10^−4^	9.65 × 10^−4^
	5	1	Cytokine–cytokine receptor interaction—Mus musculus (mouse)	KEGG	1.87 × 10^−4^	1.33 × 10^−3^
	5	1	B cell receptor signaling pathway	Wikipathways	2.82 × 10^−4^	1.76 × 10^−3^
	5	1	B cell receptor signaling pathway–Mus musculus (mouse)	KEGG	7.17 × 10^−4^	3.97 × 10^−3^
	5	1	Regulation of complement cascade	Reactome	1.03 × 10^−3^	5.40 × 10^−3^
	5	1	Complement cascade	Reactome	1.79 × 10^−3^	8.85 × 10^−3^
	5	1	Complement and coagulation cascades	Wikipathways	2.50 × 10^−3^	1.07 × 10^−2^
	5	1	Signaling by the B cell receptor (BCR)	Reactome	2.99 × 10^−3^	1.22 × 10^−2^
	5	1	TNF receptor superfamily (TNFSF) members mediating non-canonical NF-kB pathway	Reactome	3.44 × 10^−3^	1.35 × 10^−2^
	5	1	Costimulation by the CD28 family	Reactome	5.31 × 10^−3^	1.85 × 10^−2^
	5	1	Complement and coagulation cascades–Mus musculus (mouse)	KEGG	6.08 × 10^−3^	1.98 × 10^−2^
	5	1	Innate immune system	Reactome	6.10 × 10^−3^	1.98 × 10^−2^
Signaling	5	1	Olfactory transduction–Mus musculus (mouse)	KEGG	6.64 × 10^−8^	6.24 × 10^−6^
	5	1	GPCR downstream signaling	Reactome	2.39 × 10^−5^	4.50 × 10^−4^
	5	1	Signaling by GPCR	Reactome	3.42 × 10^−5^	5.15 × 10^−4^
	5	1	Olfactory signaling pathway	Reactome	3.84 × 10^−5^	5.15 × 10^−4^
	5	1	Peptide-ligand-binding receptors	Reactome	9.38 × 10^−5^	8.82 × 10^−4^
	5	1	GPCR ligand binding	Reactome	1.23 × 10^−4^	9.65 × 10^−4^
	5	1	Class A/1 (rhodopsin-like receptors)	Reactome	1.98 × 10^−4^	1.33 × 10^−3^
	5	1	G alpha (s) signaling events	Reactome	3.46 × 10^−4^	2.03 × 10^−3^
	5	1	Peptide GPCRs	Wikipathways	2.20 × 10^−3^	9.83 × 10^−3^
	5	1	Signal transduction	Reactome	3.64 × 10^−3^	1.37 × 10^−2^
	5	1	GPCRs, other	Wikipathways	5.06 × 10^−3^	1.83 × 10^−2^
	5	1	Generation of second messenger molecules	Reactome	8.05 × 10^−3^	2.52 × 10^−2^
Gene expression	5	2	Regulation of TP53 activity through phosphorylation	Reactome	1.07 × 10^−4^	3.70 × 10^−4^
Cell cycle	5	2	Cell cycle	Reactome	1.25 × 10^−24^	1.69 × 10^−22^
	5	2	Cell cycle, mitotic	Reactome	1.70 × 10^−23^	1.14 × 10^−21^
	5	2	Resolution of sister chromatid cohesion	Reactome	3.13 × 10^−23^	1.41 × 10^−21^
	5	2	Mitotic prometaphase	Reactome	3.49 × 10^−22^	1.18 × 10^−20^
	5	2	Cell cycle checkpoints	Reactome	6.95 × 10^−21^	1.88 × 10^−19^
	5	2	M phase	Reactome	6.48 × 10^−^^19^	1.46 × 10^−17^
	5	2	Amplification of signal from unattached kinetochores via a MAD2 inhibitory signal	Reactome	2.53 × 10^−17^	4.27 × 10^−16^
	5	2	Amplification of signal from the kinetochores	Reactome	2.53 × 10^−17^	4.27 × 10^−16^
	5	2	Separation of sister chromatids	Reactome	4.59 × 10^−17^	6.88 × 10^−16^
	5	2	Mitotic anaphase	Reactome	6.62 × 10^−17^	8.94 × 10^−16^
	5	2	Mitotic metaphase and anaphase	Reactome	7.46 × 10^−17^	9.16 × 10^−16^
	5	2	Mitotic spindle checkpoint	Reactome	2.52 × 10^−16^	2.83 × 10^−15^
	5	2	G2/M transition	Reactome	5.16 × 10^−10^	4.35 × 10^−9^
	5	2	Mitotic G2-G2/M phases	Reactome	6.08 × 10^−10^	4.82 × 10^−9^
	5	2	APC/C-mediated degradation of cell cycle proteins	Reactome	1.25 × 10^−9^	8.86 × 10^−9^
	5	2	Regulation of mitotic cell cycle	Reactome	1.25 × 10^−9^	8.86 × 10^−9^
	5	2	The role of GTSE1 in G2/M progression after G2 checkpoint	Reactome	1.54 × 10^−9^	1.04 × 10^−8^
	5	2	Cyclin A/B1/B2-associated events during G2/M transition	Reactome	2.04 × 10^−9^	1.31 × 10^−8^
	5	2	Activation of NIMA kinases NEK9, NEK6, and NEK7	Reactome	2.42 × 10^−8^	1.48 × 10^−7^
	5	2	Activation of APC/C- and APC/C:Cdc20-mediated degradation of mitotic proteins	Reactome	4.40 × 10^−7^	2.48 × 10^−6^
	5	2	G2/M DNA replication checkpoint	Reactome	1.39 × 10^−6^	7.22 × 10^−6^
	5	2	Cell cycle–Mus musculus (mouse)	KEGG	2.43 × 10^−6^	1.22 × 10^−5^
	5	2	Phosphorylation of Emi1	Reactome	2.77 × 10^−6^	1.34 × 10^−5^
	5	2	Regulation of PLK1 activity at G2/M transition	Reactome	4.21 × 10^−6^	1.96 × 10^−5^
	5	2	Cell cycle	Wikipathways	4.52 × 10^−6^	2.03 × 10^−5^
	5	2	TP53-regulated transcription of genes Involved in G2 cell cycle arrest	Reactome	4.83 × 10^−6^	2.10 × 10^−5^
	5	2	APC/C:Cdc20-mediated degradation of mitotic proteins	Reactome	1.51 × 10^−5^	6.35 × 10^−5^
	5	2	TP53-regulates transcription of cell cycle genes	Reactome	2.25 × 10^−5^	9.22 × 10^−5^
	5	2	AURKA activation by TPX2	Reactome	2.97 × 10^−5^	1.18 × 10^−4^
	5	2	Regulation of APC/C activators between G1/S and early anaphase	Reactome	3.24 × 10^−5^	1.25 × 10^−4^
	5	2	Condensation of prophase chromosomes	Reactome	3.86 × 10^−5^	1.45 × 10^−4^
	5	2	G2/M checkpoints	Reactome	7.26 × 10^−5^	2.65 × 10^−4^
	5	2	Nuclear envelope breakdown	Reactome	8.12 × 10^−5^	2.88 × 10^−4^
	5	2	Phosphorylation of the APC/C	Reactome	1.28 × 10^−4^	4.32 × 10^−4^
	5	2	Regulation of TP53 activity	Reactome	1.46 × 10^−4^	4.80 × 10^−4^
	5	2	Polo-like-kinase-mediated events	Reactome	2.72 × 10^−4^	8.76 × 10^−4^
	5	2	APC/C:Cdh1-mediated degradation of Cdc20 and other APC/C:Cdh1-targeted proteins in late mitosis/early G1	Reactome	2.97 × 10^−4^	8.91 × 10^−4^
	5	2	Deposition of new CENPA-containing nucleosomes at the centromere	Reactome	2.97 × 10^−4^	8.91 × 10^−4^
	5	2	Nucleosome assembly	Reactome	2.97 × 10^−4^	8.91 × 10^−4^
	5	2	p53 signaling	Wikipathways	3.73 × 10^−4^	1.08 × 10^−3^
	5	2	APC:Cdc20-mediated degradation of cell cycle proteins prior to satisfation of the cell cycle checkpoint	Reactome	3.75 × 10^−4^	1.08 × 10^−3^
	5	2	Mitotic prophase	Reactome	4.19 × 10^−4^	1.18 × 10^−3^
	5	2	p53 signaling pathway–Mus musculus (mouse)	KEGG	4.69 × 10^−4^	1.28 × 10^−3^
	5	2	Transcriptional regulation by TP53	Reactome	4.76 × 10^−4^	1.28 × 10^−3^
	5	2	Nuclear pore complex (NPC) disassembly	Reactome	7.47 × 10^−4^	1.96 × 10^−3^
	5	2	Establishment of sister chromatid cohesion	Reactome	1.47 × 10^−3^	3.48 × 10^−3^
	5	2	Depolymerisation of the nuclear lamina	Reactome	1.47 × 10^−3^	3.48 × 10^−3^
	5	2	Chk1/Chk2(Cds1)-mediated inactivation of the cyclin B:Cdk1 complex	Reactome	1.76 × 10^−3^	4.09 × 10^−3^
	5	2	Chromosome maintenance	Reactome	2.44 × 10^−3^	5.58 × 10^−3^
	5	2	G2/M DNA damage checkpoint	Reactome	3.70 × 10^−3^	8.32 × 10^−3^
	5	2	Transcriptional regulation by RUNX2	Reactome	3.99 × 10^−3^	8.83 × 10^−3^
	5	2	Loss of Nlp from mitotic centrosomes	Reactome	5.07 × 10^−3^	1.07 × 10^−2^
	5	2	Loss of proteins required for interphase microtubule organization from the centrosome	Reactome	5.07 × 10^−3^	1.07 × 10^−2^
	5	2	Hedgehog signaling pathway	Wikipathways	5.42 × 10^−3^	1.13 × 10^−2^
	5	2	APC/C:Cdc20-mediated degradation of cyclin B	Reactome	5.94 × 10^−3^	1.22 × 10^−2^
	5	2	HDR through homologous recombination (HR) or single-strand annealing (SSA)	Reactome	6.97 × 10^−3^	1.36 × 10^−2^
	5	2	Recruitment of mitotic centrosome proteins and complexes	Reactome	6.97 × 10^−3^	1.36 × 10^−2^
	5	2	Centrosome maturation	Reactome	6.97 × 10^−3^	1.36 × 10^−2^
	5	2	Recruitment of NuMA to mitotic centrosomes	Reactome	7.24 × 10^−3^	1.40 × 10^−2^
	5	2	COPI-dependent Golgi-to-ER retrograde traffic	Reactome	8.06 × 10^−3^	1.53 × 10^−2^
	5	2	Cdc20:Phospho-APC/C-mediated degradation of Cyclin A	Reactome	8.25 × 10^−3^	1.55 × 10^−2^
Development	5	2	Progesterone-mediated oocyte maturation–Mus musculus (mouse)	KEGG	2.15 × 10^−7^	1.26 × 10^−6^
	5	2	Oocyte meiosis–Mus musculus (mouse)	KEGG	7.86 × 10^−7^	4.24 × 10^−6^
DNA damage	5	2	miRNA regulation of DNA damage response	Wikipathways	4.06 × 10^−3^	8.85 × 10^−3^
	5	2	Homology-directed repair	Reactome	8.64 × 10^−3^	1.60 × 10^−2^
Haemostasis	5	2	Kinesins	Reactome	1.12 × 10^−3^	2.80 × 10^−3^
Senescence	5	2	Cellular senescence–Mus musculus (mouse)	KEGG	1.31 × 10^−3^	3.22 × 10^−3^
Signaling	5	2	RHO GTPase effectors	Reactome	5.91 × 10^−16^	5.93 × 10^−15^
	5	2	RHO GTPases-activated formins	Reactome	6.15 × 10^−16^	5.93 × 10^−15^
	5	2	Signaling by RHO GTPases	Reactome	7.96 × 10^−13^	7.17 × 10^−12^
	5	2	RHO-GTPases-activated CIT	Reactome	7.55 × 10^−4^	1.96 × 10^−3^
	5	2	Signal transduction	Reactome	8.16 × 10^−4^	2.08 × 10^−3^
Extracellular matrix organization	5	3	Activation of matrix metalloproteinases	Reactome	9.91 × 10^−3^	5.25 × 10^−2^
Haemostasis	5	3	Blood clotting cascade	Wikipathways	1.35 × 10^−3^	1.02 × 10^−2^
	5	3	Dissolution of fibrin clot	Reactome	3.52 × 10^−3^	2.34 × 10^−2^
	5	3	Response to elevated platelet cytosolic Ca^2+^	Reactome	5.05 × 10^−3^	2.97 × 10^−2^
Signaling	5	3	Olfactory transduction–Mus musculus (mouse)	KEGG	1.64 × 10^−26^	8.69 × 10^−25^
	5	3	Olfactory signaling pathway	Reactome	2.26 × 10^−9^	5.99 × 10^−8^
	5	3	G alpha (s) signaling events	Reactome	2.66 × 10^−6^	4.70 × 10^−5^
	5	3	Signaling by GPCR	Reactome	8.96 × 10^−5^	1.19 × 10^−3^
	5	3	GPCR downstream signaling	Reactome	2.94 × 10^−4^	2.70 × 10^−3^
Transport	5	3	Transport of fatty acids	Reactome	3.06 × 10^−4^	2.70 × 10^−3^
Immune response	5	4	Inflammatory mediator regulation of TRP channels–Mus musculus (mouse)	KEGG	6.85 × 10^−3^	2.74 × 10^−2^

## Data Availability

Normalized data available at NCBI https://www.ncbi.nlm.nih.gov/geo/query/acc.cgi?acc=GSE186727, access token: otenqamsthwdfep.

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
