# Peer review of "The Effects of the Food Additive Titanium Dioxide (E171) on Tumor Formation and Gene Expression in the Colon of a Transgenic Mouse Model for Colorectal Cancer"

_nanomaterials, 2022, doi:10.3390/nano12081256_

Round 1

Reviewer 1 Report

The Authors have sufficiently addressed the concerns of the reviewers.

Author Response

Dear reviewer,

We thank you for taking you time reading the manuscript and for the valuable feedback that you have provide to improve this submission.

Best regards,

Nicolaj Bischoff

Reviewer 2 Report

In this study, the authors showed the potential of E171 to induce tumor formation and progression in a CACTg/Tg;APC580S/+ transgenic mouse model. And they used ORA and STEM analyses to identify significantly modulated genes in genetic pathways and their temporality, to define mechanisms that are potentially involved in the increased tumor formation observed after the exposure to E171. Although this article was well-written, it is not suitable to be published in this journal in its current form. The main reasons are listed below.

1) To illustrate whether E171 could induce tumor formation and progression in a CACTg/Tg;APC580S/+ transgenic mouse model, the experiment groups should include a positive control besides a negative control.

Author Response

Dear esteemed reviewer, we thank you for assessing our manuscript and providing your input on this matter. We addressed your concerns and provide our response below:

  • To illustrate whether E171 could induce tumor formation and progression in a CACTg/Tg;APC580S/+ transgenic mouse model, the experiment groups should include a positive control besides a negative control.

Previous experiments in our group and with our collaborators explored the effects of E171 in a colitis induced mouse model, utilizing an DSS/AOM induced colitis model for colorectal cancer. The experiments showed that DSS/AOM + E171 significantly increased the number of tumors in this model (Urrutia-Ortega et al. 2016). Since increased tumor formation in this mouse model was observed after exposure to E171 in combination with DSS and AOM, we wanted to investigate the effects of E171 in a mouse model independent of the tumor formation by DSS/AOM. Hence, we decided to investigate this in the transgenic mouse model of Xue et al. (2010), which provides an environment which mimics the human situation of sporadic colon cancer, which is prone to the development of colorectal cancer within the first 10 weeks of age, and independent of chemical induction. The aim of our functional study was not to identify E171 as a carcinogen itself, but rather to investigate the effects of E171 on stimulation or acceleration the development of colorectal cancer, as observed in our previous experiments in the colitis associated cancer model. In the tumor formation study in the transgenic model no significant changes on either of these parameters were found, despite the indication of increased tumor volume, as described in results of our pilot study. In this experimental set-up an adequate vehicle and administration control was included. The addition of a known carcinogen (positive control) to a tumor prone environment, does not provide any additional information or validation in this Tg mouse model from Xue at al. (2010). We acknowledge that the tumor incidence within our control group is larger than previously described by Xue et al. (2010), which might be due to a smaller sample size, or to the different route of administration. Our study design relied on the intragastric administration (gavage) of E171/ vehicle control, in comparison to drinking water in the study by Xue et al. (2010)/ pilot study.

The main objective of this study was to investigate whole gene expression changes to identify molecular processes induced by E171 that precedes tumor formation. We designed the experiments in such a way that would allow the comparison to previous studies, with a similar experiment set up, as published by Proquin et al. (2017, 2018), to identify gene expression changes that might help to explain the possible contribution of E171 in tumor formation, as described in the literature (Bettini et al. 2017, Urrutia-Ortega et al 2016). To support these findings, we tried to investigate further markers, such as histopathological changes in the colon tissue/adenocarcinomas (pilot study), as well as the assessment of other body parameters after E171 exposures e.g., body weight, organ weight and the number of tumors found in comparison to the unexposed negative control (tumor formation study).

The addition of a positive control for a gene expression study for E171 is difficult, since such a control would need to mimic the exact molecular mechanisms as the substance that we would like to investigate. Such a selection of an appropriate is in our opinion hardly possible. The same positive control would need to be deployed for the tumor formation study, to guarantee a coherent experimental design. We therefore argue that the experimental set-up we have chosen is adequate to describe the gene expression changes in early stage colorectal tumors that are developed in a tumor prone environment, which mimics the initiation of sporadic colorectal cancer in such a way that this is more comparable to that in humans, and by using this design, that our study contributes to a better understanding of potential molecular mechanisms that are involved in E171 related adverse effects. 

We once more want to thank the reviewer for the critical reading and hope that we have addressed the concerns sufficiently.

Reviewer 3 Report

The article 'The effects of the food additive titanium dioxide (E171) on tumor formation and gene expression in the colon of a transgenic mouse model for colorectal cancer' is interesting read.  The doses used are physiologically relevant as indicated in the EFSA and highlighted by the authors. A range of genes have been affected subsequent to exposure by E171 which is concerning given that many NM find their way into products lacking intensive risk assessment. 

Here are few minor suggestions:

  1. Adverse effects of TiO2 were first studied after inhalation. Please provide references as this is crucial for the reader to know that based on these findings the IARC has  classified TiO2 as “possibly carcinogenic to humans after inhalation”
  2. Indicate genes in venn diagram in the legends 
  3. Fig 10 - minor comment - please place legend in the bottom right corner. 
  4. As indicated above, the authors should highlight in the conclusion the importance of detailed toxicity studies to evaluate the safety of similar products that may find their way into consumer market. 

Author Response

Dear esteemed Reviewer,

We thank you for taking your time to reassess our submission and provide valuable suggestions to improve this manuscript. We tried to address your remarks to the best of our abilities.

  • Adverse effects of TiO2 were first studied after inhalation. Please provide references as this is crucial for the reader to know that based on these findings the IARC has classified TiO2 as “possibly carcinogenic to humans after inhalation”

The studies leading to the classification as carcinogen class 2 B according to IARC are summarized in the monogram that is referenced in the introduction. To cite the all the publications leading to this classification would drastically increase the number of citations within the introduction. We have been previously advised by your colleague (reviewer 2) to reduce the number of citations within this manuscript, which we did after the first round of reviewing. 

  • Indicate genes in venn diagram in the legends.

The genes that overlap are indicated in the description of the Venn diagram, which describes those genes that overlap between 3 or more timepoints. For a more detailed overview, it is possible to use the heatmap which list all genes that are significantly differentially expressed (marked with *) per time point. Those are also grouped by the number of timepoints they are overlapping with each other.

  • Fig 10 - minor comment - please place legend in the bottom right corner. 

The legend for the visualized genetic pathway network is presented at the bottom right corner of the image.

  • As indicated above, the authors should highlight in the conclusion the importance of detailed toxicity studies to evaluate the safety of similar products that may find their way into consumer market. 

Line 828-830: Addition of the following sentence. “Further toxicity studies are needed to evaluate the safety of E171 and other metal-based nanomaterials, which are used as food additives or food packaging materials.”

We once more want to thank the reviewer for the valuable input and hope that we could address the concerns sufficient.

Round 2

Reviewer 2 Report

Our comments have been illustrated clearly and properly. The article is suitable to be published in its current form. 

This manuscript is a resubmission of an earlier submission. The following is a list of the peer review reports and author responses from that submission.

Round 1

Reviewer 1 Report

The paper describes a number of well performed animal experiments with a focus on genetic pathways. The data suggest that E171 may  contribute to tumor formation and progression by modulation of pathways related to inflammation, activation of immune responses, cell cycle events, and cancer signaling.  These findings can be of relevance in the ongoing debate on the safety evaluation of E171.

Page 2 line 89. Further description of the transgenic mice model is needed. The APC and CAC gene is mentioned. Give more information on the expression of this and other genes

Was the model used in other studies on chemically induced tumors in the colon? Give som references.

What is Cre-Lox P system?

Line 113: Is the E171 food grade quality. Can you provide a particle size distribution. Does the sonication influence the particle size distribution?

Line 141: 11µ/ml. Are we missing something after the micro symbol? How was the dose selected. What was the numner of particles in the dosing solution and how does this compare with the levels E171 in food products.

Line 159: what was the number of particles in the dosing solution. How was this dose selected? How does the doses related to human exposure?

Line 167: The mice with rectal prolapse: was the tumor burden studied in these mice+

Line 171: what was the age of the mice when the treatment started.

Line 180: how much of the colon was sampled. How many parts of the liver were sampled?

Line 184: describe in more detail (length  of the colon collected, distance to the anus). Was the clonon rinsed before the treatment with the beater. Were there any macroscopic observations?

Line 282: The mice were exposed… This sentence is incomplete

Line 289: Did you observe any E171 particles in the tissues?

Line 305: add a legend to both figures.

Why is the spleen weight  figure included twice. Where is the figure for the liver?

Line 249: was histology performed on these tumors and were the findings comparable to the pilot study. What about the size of the tumors.

Line 396: suggest to explain DEG in the legend text.

Line 600: since the findings described here didn’t achieve statistical significance, I suggest using the words trends

Line 619: low-grade inflammation and increased development of tumors and their size: please indicate more clearly that this is a hypothesis. Are there any other chemicals that are speculated to have the same mechanisms? Is there an AOP that can be referred to.

Reviewer 2 Report

In this study, the authors showed the potential of E171 to induce tumor formation and progression in a CACTg/Tg;APC580S/+ transgenic mouse model. And they used ORA and STEM analyses to identify significantly modulated genes in genetic pathways and their temporality, to define mechanisms that are potentially involved in the increased tumor formation observed after the exposure to E171. Although this article was well-written, it is not suitable to be published in this journal in its current form. The main reasons are listed below.

1) The novelty of this articles differed from published articles should be stated clearly.

2) There are too many references.

Reviewer 3 Report

The Authors have submitted a comprehensive investigation regarding the potential E171-induced genotoxicity and carcinogenicity.

The manuscript is interesting, timely, and addresses a topic of high significance. The employment of transgenic mouse models is of particular interest. On the other hand, because the associated potential impact of these findings on the scientific field and even more on the society, the findings have to be discussed with a special carefulness. Overall, I suggest its resubmission only after the following majors have been fully addressed:

- It is of critical importance to do not mix info regarding TiO2 and to carefully specify if associated to the bulk, micro or nano form because size matter. Thus, Authors should modify many points in the maintext, as for example lines 55-63. A summarizing table reporting the actual (or absent) data or classifications for bulk, micro or nano TiO2 should be reported in the intro.

- Authors sonicate E171 before the experiments. TEM images of E171 after the 30 mins sonication should be reported to confirm that the size/shape have been not modified after the strong mechanical stimulation.

- It is pivotal to explain why the Authors have chosen to treat the models with 1 or 5 mg/kgbw/day. Is it a relevant dose? Is it the average humans ingested E171 allometrically scaled for mice?

- Fig. 3A-B: Authors state the average mice with tumors and tumors number/mice is increased after the treatment with E171. On the other hand, it seems there are not statistically significant differences. Thus, the Authors should modify all the manuscript, the title and the abstract clearly reporting that “a not-significant increase in the number of tumors and in the number of mice treated with E171 have been observed”.

- Lines 636: “we hypothesized that E171 exposure enhances colorectal cancer development”. This statement is simply not true by looking at the point before.

- Fig. 4: where are the error bars? It is not possible to speculate any tendency without significance analysis. Thus, the sentences in lines 627-638 are speculation not based on solid data.

- In general, the conclusions as well as the title have to be revisited in function of the data analysis.

-As a minor, some typos have to be double checked, as for example the missed “reported” in line 98.